# Measuring hierarchical structure across hominid percussive tool-use sequences
Derry Taylor [1,2] ✉, Tina Petersen[1,3], Catherine Crockford[1,2,4] & Roman M. Wittig [1,2,4] ✉

Understanding the evolution of animal cognitive capacities requires us to study their full range of naturally occurring sequences of behavior. It has long been theorized that cognitive capacities are revealed through the sequential structure of natural behavior, particularly its hierarchical organization. Progress in understanding the origins of this capacity has, however, been limited by a lack of techniques for identifying and measuring hierarchical structure in behavioral sequences. To fill this methodological gap, we introduce here an analysis pipeline for measuring hierarchical structure in sequential behavior. We then establish the validity of our approach by first applying it to chimpanzee percussive tool-use (PTU) sequences and comparing it to markov-simulated control sequences. Secondly, we apply our analysis to a dataset on PTU in humans and compare the hierarchical complexity of chimpanzee and human PTU. Despite decades of speculation, our study is the first empirical demonstration of hierarchical structuring in chimpanzee tool-use. We found chimpanzee PTU is characterized by a level of hierarchical complexity beyond that which can be generated through markov process, but is nonetheless systematically less hierarchically complex than human PTU, as expected. Altogether, our analyses demonstrate the potential for our approach to successfully detect and measure hierarchical structuring in natural sequences of behavior, which we believe will play a pivotal role in shedding light on old questions, as well as opening up entirely new lines of inquiry in the study of human and animal behavior.

The hierarchical organization of sequential behavior is central to many hallmark features of human cognition[1]. Human language, for example, is a hierarchically structured system, with sequences of phonemes organized into morphemes, which are in turn organized into words, and then phrases of potentially unbounded complexity[2]. Tool-making and tool-use sequences also display hierarchical structure, and many authors argue that these capacities reflect evolutionary precursors of the hierarchical cognition that supports language[3,4]. On this view, the ability to represent and execute hierarchically organized action is not uniquely human but shared with great apes, with human syntax representing a quantitative elaboration of these capacities rather than an entirely novel faculty. In support of this hypothesis, several studies have claimed that chimpanzee tool-use is hierarchically organized[5–7]. However, there are currently no standardized techniques for systematically identifying and quantifying hierarchical structure in natural sequences of behavior. Unsurprisingly, then, the claim that language and action share any similarity in organization remains contested[8].

Therefore, we built an analysis pipeline as an extension of the hierarchical compression algorithm "Sequitur", developed by ref. 9 which processes linear sequences and outputs a compressed version of the original sequence and a rule-book which allows the original sequence to be reconstructed. Indeed, the sequitur algorithm has been used to measure *compressibility* (i.e., the degree of repetitiveness) of behavioral sequences across several species[10,11]. However, equal compressibility can be achieved through different means, some of which may not be hierarchical at all, depending on sequence structure (i.e., through the formation of many non-hierarchically embedded rules vs fewer but more hierarchically complex rules, see Fig. 1). Further, for any given sequence, there are many different possible ways that it may be hierarchically organized, and this space of possibilities grows exponentially as a function of sequence length (Stanley[12], see Fig. 2). This is important not to overlook because different forms of hierarchical organization correspond to different forms of grammar and are not therefore computationally identical[13]. Direct and transparent measurements are needed to better understand hierarchical structure in sequential behavior, make comparisons across domains and across species, and draw inferences about its cognitive implications. As such, we decided to build an analysis pipeline that extracts more transparent measurements from sequitur rule-books, better

[1]Ape Social Mind Lab, Institute des Sciences Cognitives Marc Jeannerod, University Lyon 1, Bron, France. [2]Evolution of Brain Connectivity Project, MPI for Evolutionary Anthropology, Leipzig, Germany. [3]School of Psychology and Neuroscience, University of St Andrews, St Andrews, Scotland, UK. [4]Taï Chimpanzee Project, Centre Suisse de Recherche Scientifique, Abidjan, Côte d'Ivoire. ✉e-mail: dtaylor@isc.cnrs.fr; rwittig@isc.cnrs.fr

**Fig. 1 | An illustration of how compressibility using hierarchical algorithms such as sequitur does not necessarily correspond to hierarchical structure.** Sequence A and B are of identical length (12 characters) and are both equally compressible (i.e., to 6 characters). The top panel shows the original sequence. The middle panel shows the compressed sequence (R0) and the rules found within the sequence. Using R0 and the other rules, the original sequence can be completely reconstructed. For example, R0 for sequence A begins with R1. R1 corresponds to the bigram "a a". Correspondingly, the first pair of characters in the original sequence is "a a". However, sequence A contains no hierarchical structuring (that is, rules embedded within rules). By contrast, sequence B does contain hierarchical structuring (R1 is embedded within R2). This is due to difference in sequence structure. This clearly demonstrates that compressibility does not correspond to hierarchical structure, which needs to be measured directly.

| | Sequence A | Sequence B |
|---|---|---|
| **Original sequence** | a a a a b b b b c c c c | a b a b c a b c d a b f |
| **Compressed sequence (R0) and rules** | R0 -> R1 R1 R2 R2 R3 R3<br>R1 -> a a<br>R2 -> b b<br>R3 -> c c | R0 -> R1 R2 R2 d R1 f<br>R1 -> a b<br>R2 -> R1 c |
| **Representation of hierarchical structure** | | |

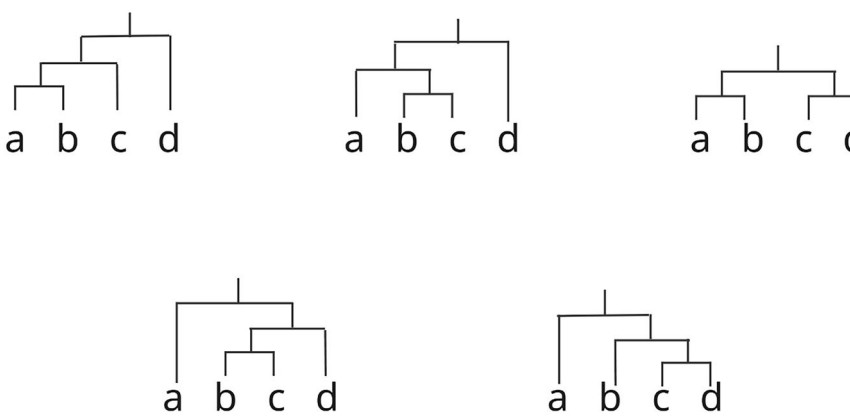

**Fig. 2 | An illustration of the variety of ways in which the same sequence can be hierarchically organised.** Hierarchical structure is latent, meaning it is not directly observable in a linear sequence (e.g., a -> b -> c -> d). A priori, there are many possible hierarchicaly structures that *could* be mapped onto a given sequence, which grows exponentially with sequence length.

suited for testing hypotheses about the origins of hierarchical structuring in human behavior to provide greater insights into sequence complexity.

It has been formally demonstrated that hierarchical structuring cannot be generated by Markov processes, since Markov processes are by definition memoryless, and hierarchical structuring involves dependences between non-adjacent elements in a sequence that go beyond a Markov chain's capabilities[14]. As a proof of concept, we our applied our analysis pipeline to PTU in chimpanzees and humans, focusing on: (i) nut-cracking in chimpanzees[15] and (ii) stone-tool production in humans[16]. We first establish the presence of hierarchical structuring in chimpanzee PTU by combining the probabilistic (i.e., Markov modeling) and deterministic (i.e., information compression) approaches used in parallel in previous studies on this topic[16]. Specifically, we compared hierarchical complexity in real chimpanzee tool-use sequences to Markov-simulated control sequences, generated by a Markov model trained on chimpanzee PTU sequences. We then compare the hierarchical complexity of chimpanzee PTU to a system with known hierarchical structure—hominin percussive stone-tool manufacturing[16], hypothesized by some to be a direct elaboration on the percussive abilities of ancestral primates[17], finding that chimpanzee PTU shows systematically lower hierarchical complexity, demonstrating the potential to quantify the extent of this difference for the first time.

## Results
### Chimpanzee PTU sequence structure and diversity
Compared to a randomly generated and markov simulated control sequences we found that chimpanzees typically had rules with a greater phrase lengths compared to both random and markov sequences (chimpanzees: $\alpha = 1.05$, 95% CI-[1.01, 1.09]; random: $\alpha = -0.16$, 95% CI [−0.22, −0.09]; markov: $\alpha = -0.07$, 95% CI [−0.13, −0.01] Fig. 3c). Not only were chimpanzee phrases longer, they were also more complex than both random and markov sequences, as measured by the entropy of tree structures (chimpanzees: $\alpha = 1.94$, 95% CI [1.88, 2.00]; random: $\alpha = -0.29$, 95% CI [−0.38, −0.20]; markov: $\alpha = -0.19$, 95% CI [−0.28, −0.11] Fig. 3d). Chimpanzee tree structures had a higher gini index than markov but not random sequences, indicating a more even distribution of non-terminal nodes across trees (chimpanzees: $\alpha = 0.98$, 95% CI [1.04, 0.91]; random: $\alpha = -0.03$, 95% CI [−0.15, 0.10]; markov: $\alpha = -0.13$, 95% CI [−0.23, −0.02] Fig. 3e). Finally, chimpanzee tree structures showed a higher terminal:non-terminal ratio compared to markov and random sequence (chimpanzees: $\alpha = -0.25$, 95% CI [−0.30, −0.21]; random: $\alpha = -0.22$, 95% CI [−0.29, −0.14]; Markov: $\alpha = -0.14$, 95% CI [−0.21, −0.08] Fig. 3f). Chimpanzee sequences were more complex in terms of hierarchical depth than random but not markov sequence rules (chimpanzees: $\alpha = 0.73$, 95% CI [0.61, 0.85]; random: $\alpha = -0.25$, 95% CI [−0.44, −0.06]; Markov: $\alpha = -0.07$, 95% CI [−0.25, 0.12], Fig. 3a) and chimpanzee sequences contained more rules than random but not markov sequences (chimpanzees: $\alpha = 1.76$, 95% CI [1.68, 1.84]; random: $\alpha = -0.22$, 95% CI [−0.33, −0.10]; markov: $\alpha = -0.01$, 95% CI [−0.12, 0.10] Altogether, our results show that the hierarchical complexity of chimpanzee nut-cracking sequences goes beyond that which can be produced through random or markov processes.

As the hierarchical complexity of sequences increases, the range of possible structures underlying that complexity increases exponentially. To

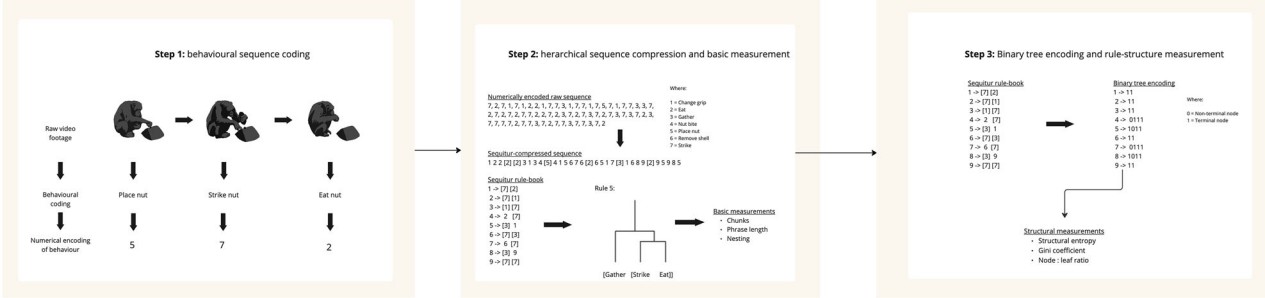

**Fig. 3 | Flow diagram depicting the data processing pipeline for extracting measurements of hierarchical complexity from behavioral sequences.** The first step involves coding behavioral sequences. Starting with video footage of the behavior of interest, in this case nut cracking, an observer manually codes the sequence of behaviors. These behaviors are then encoded numerically, with one number representing one type of behavior. At step 2, the raw numerical sequence is run through the hierarchical compression algorithm, "sequitur", which outputs a compressed version of the sequence, where each number now represents a rule (unless it is in square brackets, in which case the behavior was not compressible) and the rule-book can be used in combination with the compressed sequence to reconstruct the original sequence. Each rule in the rule-book corresponds to one "chunk", that is, one repeating subsequence from the raw sequence. Each chunk has a corresponding "phrase-length", that is, the number actions in the sub-sequence. Also, each chunk may have other chunks embedded within it. In the example above, rule 5 is one chunk, with a phrase length of 3, which is achieved by embedding one chunk (i.e., strike, eat) within a longer chunk. To extract measurements of rule structure, further processing is needed, as illustrated in step 3. First, each rule needs to be encoded in binary. The encoding is generated by reading the rule from left to right and replacing terminal nodes with '1' and non-terminal nodes with "0". From the binary tree encoding, measurements of rule structure, such as entropy (i.e., structural predictability), gini coefficient (i.e., evenness of structure in terms of the distribution of non-terminal nodes) and node : leaf ratio (i.e., branching complexity).

further examine the extent to which chimpanzee PTU sequence complexity goes beyond what can be captured by random or Markov-simulated controls, we examined the extent to which rule structures were shared between chimpanzee, Markov-simulated, and random sequences. After removing all repeated patterns, there were only 4 unique rule structures within the random sequences, all of which were shared with Markov and chimpanzee sequences. There were 20 unique rule structures in our Markov sequences, only 1 of which was not shared with chimpanzee or random sequences. For the chimpanzees, we observed 29 unique rule structures, of which 10 were not shared with Markov-simulated or randomly generated sequences. To better understand why chimpanzee sequences contain a greater number of unique structures, we compared the structure and complexity of shared vs unshared (i.e., unique) rule structures for the chimpanzee sequences. We found that unique rules had longer pattern lengths (shared: $\alpha = 1.99$, 95% CI [1.82, 2.16]; unique: $\alpha = 0.34$, 95% CI [0.09, 0.59], Fig. 4b), higher entropy (shared: $\alpha = 0.61$, 95% CI [0.54, 0.67]; unique: $\alpha = 0.11$, 95% CI [0.01, 0.22], Fig. 4c), higher gini coefficients (shared: $\alpha = -0.59$, 95% CI [−0.68, −0.60]; unique: $\alpha = 0.19$, 95% CI [0.04, 0.34], Fig. 4d), higher terminal:non-terminal ratios (shared: $\alpha = 0.25$, 95% CI [0.08, 0.44]; unique: $\alpha = 0.44$, 95% CI [0.12, 0.76], Fig. 4e), but were much less common overall (shared: $\alpha = 3.48$, 95% CI [3.39, 3.55]; unique: $\alpha = -3.17$, 95% CI [−3.72, −2.66], Fig. 4f). Unique rules did not show greater hierarchical depth (shared: $\alpha = 0.41$, 95% CI [0.02, 0.75]; unique: $\alpha = 0.26$, 95% CI [−0.30, −0.83], Fig. 4a). Altogether, these results demonstrate that while markov models can capture simple, common structures in chimpanzee nut-cracking sequences, real sequences contain about 30% more rules that cannot be captured by markov models, owing to their hierarchical complexity.

### Hominoid PTU compared
Compared to human stone-tool manufacturing sequences, we found that chimpanzees typically had rules with less hierarchical depth, while achulean showed more than oldowan (chimpanzees: $\alpha = -0.46$, 95% CI [−0.86, −0.05]; oldowan: $\alpha = -0.04$, 95% CI [−0.56, 0.47]; achulean: $\alpha = 1.19$, 95% CI [0.79, 1.56], Fig. 5a). Chimpanzees had fewer rules per sequence compared oldowan and achulean, but oldowan showed more than achulean (chimpanzees: $\alpha = -0.80$, 95% CI [−1.01, −0.58]; oldowan: $\alpha = 2.56$, 95% CI [2.36, 2.75]; achulean: $\alpha = 0.36$, 95% CI [0.10, 0.62], Fig. 5b). Chimpanzee rules had shorter phrase lengths compared to oldowan and achulean but achulean showed more than oldowan (chimpanzees: $\alpha = -0.24$, 95% CI [−0.35, −0.12]; oldowan: $\alpha = -0.09$, 95% CI [−0.23, 0.04]; achulean:

$\alpha = 1.29$, 95% CI [1.18, 1.38], Fig. 5c). Chimpanzee rule structures had lower entropy compared to achulean but not oldowan (chimpanzees: $\alpha = -0.23$, 95% CI [−0.41, −0.07]; oldowan: $\alpha = 0.13$, 95% CI [−0.33, 0.06]; achulean: $\alpha = 2.13$, 95% CI [1.98, 2.30], Fig. 5d). Chimpanzee rule structures did not differ from oldowan or achulean in their gini coefficients (chimpanzees: $\alpha = -0.07$, 95% CI [−0.22, 0.07]; oldowan: $\alpha = -0.07$, 95% CI [−0.24, 0.09]; achulean: $\alpha = -0.91$, 95% CI [−1.04, 0.78], Fig. 5e). Finally, chimpanzee tree structures had a lower terminal:non-terminal ratio compared to oldowan and achulean but there was no difference between oldowan and achulean (chimpanzees: $\alpha = -0.16$, 95% CI [−0.29, −0.03]; oldowan: $\alpha = -0.08$, 95% CI [−0.23, 0.07]; achulean: $\alpha = -0.09$, 95% CI [−0.21, 0.04], Fig. 5f). Our results here show that, as expected, chimpanzee PTU has a systematically lower hierarchical complexity than hominin percussive stone-tool use. This demonstrated the validity of our approach for meaningful cross-species comparisons of hierarchical sequence complexity.

## Discussion
In this paper, we aimed to develop and validate an analysis pipeline that allows, for the first time, the hierarchical complexity of natural behavioral sequences to be measured and compared across species and contexts. We demonstrated that chimpanzee tool-use sequences exhibit hierarchical structuring, as shown by the fact that the hierarchical complexity of chimpanzee PTU sequences goes beyond what can be captured by Markov processes. While recent work has shown that chimpanzee PTU sequences contain non-adjacent dependencies[18], our study is the first to demonstrate hierarchical structuring directly, since non-adjacent dependencies are necessary but not sufficient for hierarchical organization[19]. This pattern mirrors findings in linguistics, where Markov models capture many short, common sequences but fail to capture longer, more complex structures[20]. Importantly, our approach allows the depth and complexity of sequences to be quantified, enabling comparisons not only within a species across different tool-use behaviors, but also across taxa and contexts (e.g., chimpanzee nut-cracking versus hominin percussive tool-making). While our results show overlapping ranges of hierarchical complexity across chimpanzees and hominins, they also reveal differences in variability, highlighting how this method can capture nuanced aspects of sequence organization without assuming systematic interspecies differences. Importantly, our deliberate use of a relatively simple action vocabulary here gives an under-representation of possible sequences of action, meaning this is most likely an

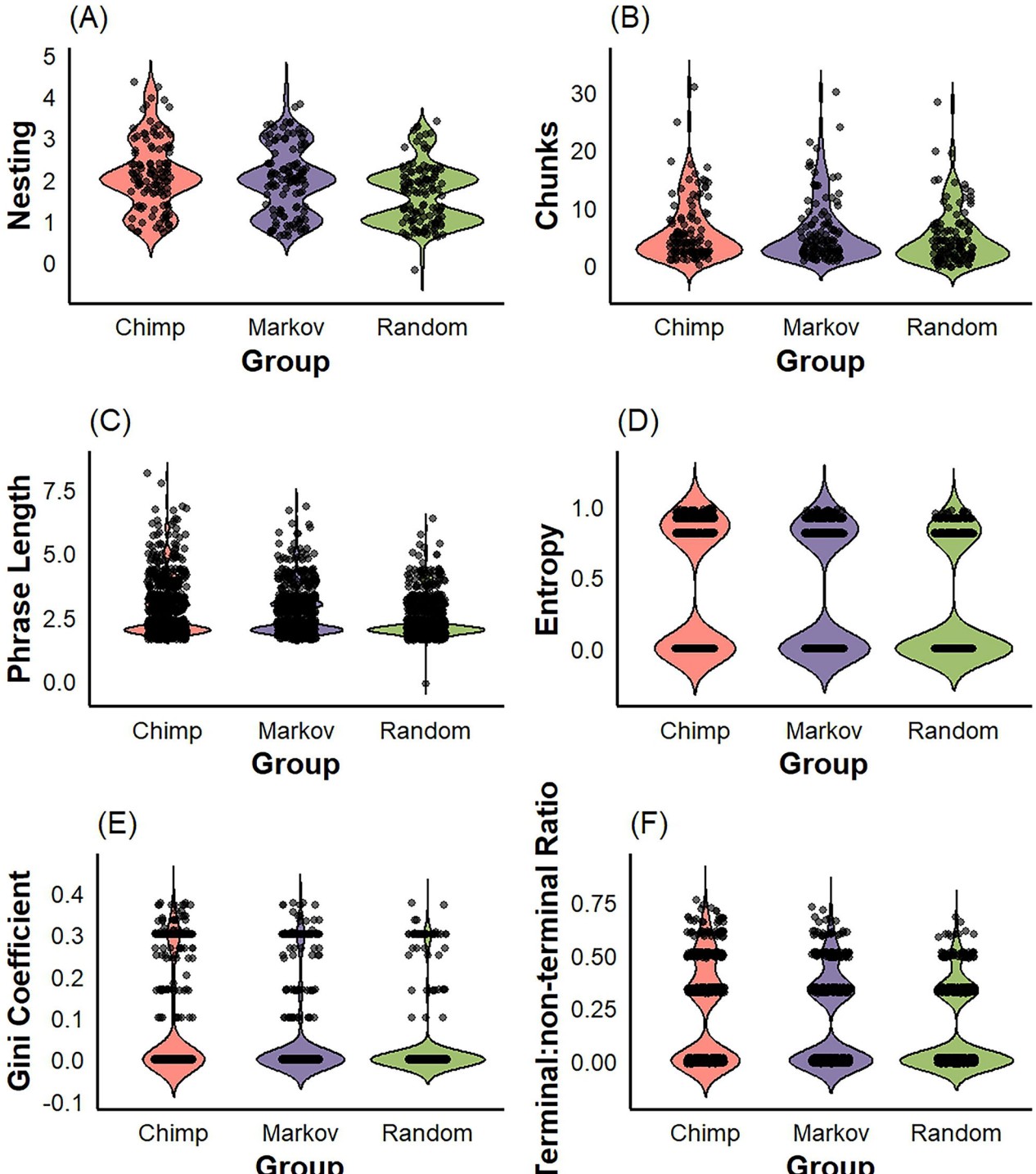

**Fig. 4 | Violin plots of hierarchical complexity of PTU action sequences in chimpanzees ($N = 140$) compared to randomly generated ($N = 140$) sequences and markov-simulated ($N = 140$) control sequences.** Black dots represent original data points and the shape around them represents kernel density estimates. **A** Hierarchical depth of chimpanzee nut-cracking rules, (**B**) Number of rules per sequence, (**C**) Length of phrases generated by rules, (**D**) Entropy of tree structures, **E** Gini coefficient of tree structures, (**F**) Terminal:non-terminal ratio of tree structures.

underestimate of the true sequential complexity in any particular domain or species studied here. Overall, these findings demonstrate that chimpanzee PTU is hierarchically structured, and that while hominin PTU overlaps considerably with what we have observed in chimpanzees regarding complexity, it seems likely that hominin PTU tool-use complexity is greater than we find in chimpanzees, illustrating the utility of our method for comparative studies across tasks and domains, and

consistent with the hypothesis of gradual evolution of hierarchical organization in percussive behaviors[17,21]

This result has direct implications for ongoing debates about whether non-human primate action sequences can be explained as associative chains of learned transitions or require representations of nested subgoals[22]. used simulations to demonstrate that chimpanzee nut-cracking, in a highly simplified form, could be learned through chaining and conditioned

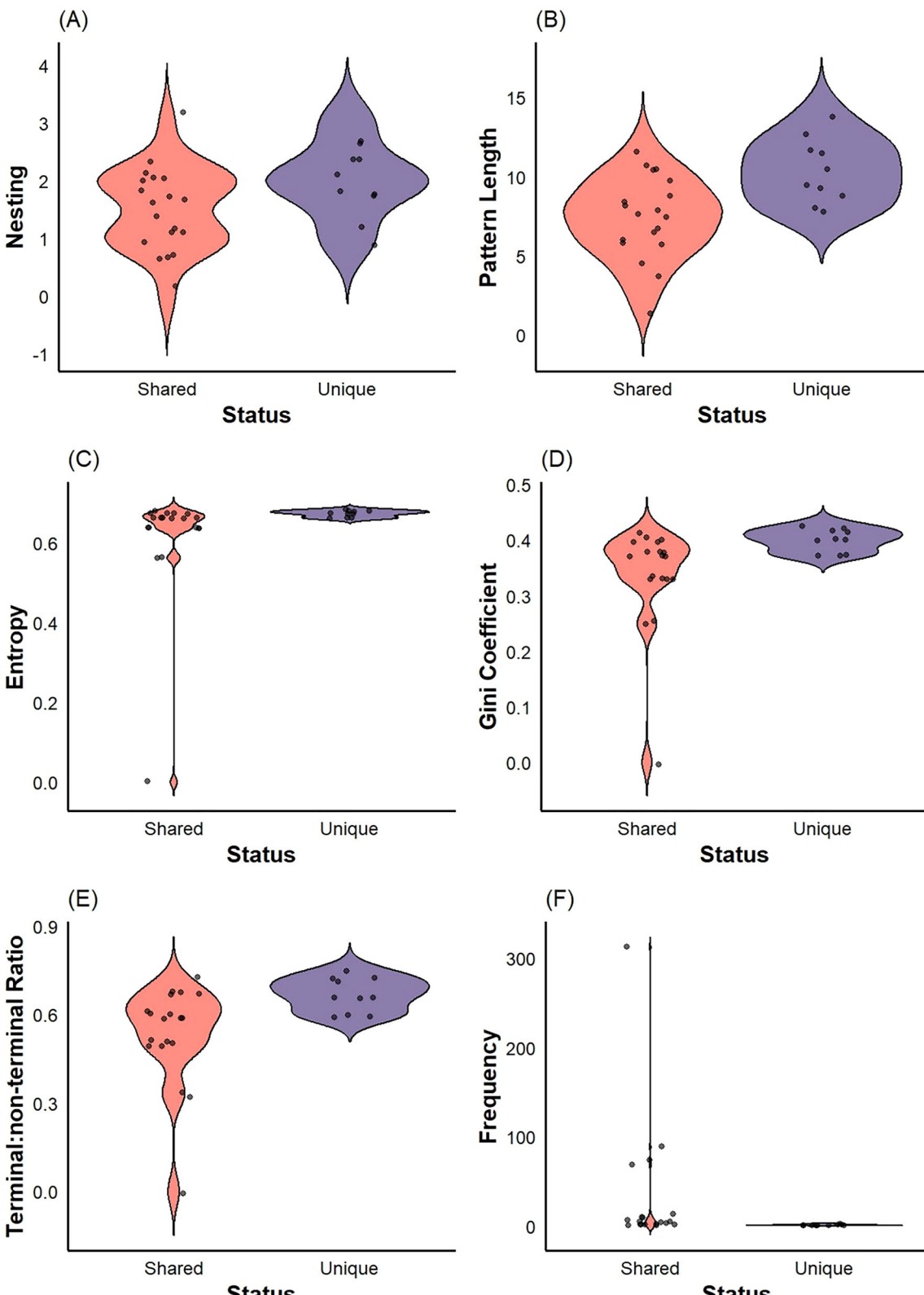

**Fig. 5 | Violin plots of hierarchical complexity of rules unique to chimpanzee PTU sequences ($N = 140$) compared to those also discovered by a markov simulation ($N = 140$) (i.e., "shared" patterns).** Black dots represent original data points and the shape around them represents kernel density estimates.

**A** Hierarchical depth of chimpanzee PTU rules, (**B**) Length of observed structures, **C** Entropy of tree structures, (**D**) Gini coefficient of tree structures, (**E**) Terminal:non-terminal ratio of tree structures, and (**F**) Frequency of shared vs unshared structures.

reinforcement. However, their approach assumes a single, fixed solution to the problem, whereas in reality nut-cracking is ecologically variable and can be solved in many different ways—for instance, an individual might first retrieve the nut or the hammer, and anvils vary greatly in their affordances, from tree roots to large rocks or even elevated branches. This diversity of solutions and ecological contexts likely imposes planning and organizational demands that go beyond simple chaining, which helps to explain why our Markov model captured much of the structure (around 70% of rules) but failed to account for the more hierarchically complex sequences. Our findings therefore complement[22]'s simulations: while chaining may suffice to explain simplified scenarios, real-world nut-cracking appears to require more flexible, hierarchical organization of action. This builds on earlier archeological work quantifying action complexity in stone tool-making[23], which argued for dynamic, innovative capacities in the Lower Palaeolithic. Our fully automated approach extends such work by directly quantifying hierarchical structure from observed sequences, offering a scalable tool for evaluating hypotheses about cognitive evolution. However, it is important to note that our analyses identify statistical signatures of hierarchy in behavior rather than revealing mental representations directly. Hierarchical structure in sequences is consistent with—but does not prove—the existence of explicit tree-like mental models, as similar structure could arise from chunking strategies, procedural routines, or biomechanical constraints. As such, our results should be taken as evidence that observed behavior exhibits hierarchical organization beyond local associations, rather than as direct evidence of the precise neural or representational mechanisms generating this structure.

While the question of hierarchical structuring in action sequences has continued to be debated[8], most have focused on formal analysis of simple and specific examples[2]. Our approach provides a set of empirical techniques for establishing the presence of hierarchical structuring in behavioral sequences, and allows us to move beyond the binary question of whether or not such structuring is present, by providing a method for quantifying the extent to which it is present. In neuroscience, it has been increasingly argued that techniques for quantifying the complexity of unconstrained behavior is essential for making progress in understanding brain function[24]. Further, it has been argued that such techniques must allow for an evolutionary approach since the brain emerges under evolutionarily inherited constraints[25]. The quantitative differences between hominoid species in the hierarchical complexity of behavior provides a context within which comparative neuroanatomical differences can be investigated[26], although it is important to note that task-specific demands may be a constraining factor on sequence structure here. A growing body of research suggests a comparative expansion in brain structures related to the production and perception of hierarchically structured action, such as Broca's area[27]. The methodological framework developed here opens up the possibility to make progress in understanding these neuroanatomical differences across species. By measuring the hierarchical complexity of behavior across species and domains, we can begin to disentangle the relationship between brain-structure and function within an evolutionary context, a central aim of contemporary neuroscience[28]. At the same time, evidence that great apes can acquire basic stone-flaking skills under intensive training[29] underscores that observed differences in behavioral complexity likely reflect an interaction between evolved capacities, ecological relevance, and cultural scaffolding, highlighting the need for comparative approaches that integrate both species-typical behaviors and experimentally elicited skills.

The approach we present is not limited to the specific cases of nut-cracking or percussive tool use, but offers a general framework for quantifying hierarchical structure and complexity in sequential behavior. Crucially, the measures we derive are agnostic to the specific motor or communicative domain, focusing instead on the structural properties of sequences themselves. This makes them inherently comparable across different species and tasks: the same computational principles of sequence organization—recursion, nesting, and re-use of sub-sequences—can be assessed, whether the behavior consists of tool-use actions, gesture strings,

or vocal signals. Such cross-domain comparability is particularly valuable because hypotheses about the evolution of language, culture, and cognition often rest on identifying domain-general capacities for structuring behavior hierarchically[26,30,31]. By grounding analysis in sequence structure rather than surface content, our method provides a unified metric for testing whether, for example, primate tool-use sequences share structural features with birdsong or human action planning[32,33]. In this way, the pipeline contributes not only to fine-grained analyses of particular behaviors but also to broader comparative and evolutionary debates about the origins of language, the cognitive basis of cultural innovation, and the general principles shaping complex sequential behavior across species[34,35].

One limitation of the present approach is that it treats sequences with additional or repeated actions as distinct structural units. However, it does not tell us what the function of these units is, and to what extend one unit is functionally distinct from another. This reflects our focus on describing the surface organization of action, without making assumptions about functional equivalence. However, it is possible that some of these structural differences correspond to adaptive responses to contextual variation (e.g., nut hardness, tool type, strike success), while others may represent distinct techniques. For example, in nut cracking, repetitions of a "strike" action might reflect the hardness or ripeness of the nut, the weight of the hammer stone, or an initial missed strike, each of which could alter the functional significance of the sequence. To determine whether structurally distinct sequences are functionally equivalent, future work would need to incorporate contextual and outcome measures, such as nut properties, tool characteristics, strike-by-strike outcomes, and efficiency metrics (e.g., number of strikes per successful opening). Such data would allow researchers to assess whether structural variability reflects adaptive adjustment, alternative strategies, or simply executional noise. Distinguishing between surface variation and functional equivalence remains a significant challenge, but addressing it will be crucial for linking structural descriptions of action to their functional interpretation.

In this work, we have successfully developed and illustrated the potential of a method to identify and measure hierarchical complexity in action sequences. Note, both datasets under-estimate complexity. For comparability, we decided to use an ethogram with a matching number of behaviors for describing human and chimpanzee behavior. This was a good way to demonstrate proof of a useful method to capture hierarchical processes, but with more detailed data, we may find a greater level of hierarchical complexity in both species. An adequate ethogram would include details that more fully capture variation in action sequences, such as greater detail on types of hand grip, specific percussive actions, and action laterality. A critical step achieved here is finding a suitable quantitative method that identifies hierarchically structured action in animal behavior, which provides a perspective on cognitive evolution, grounded in natural ecological settings. Further work adopting the approach developed here with more fine-grained data across species offers an objective comparative method across species that may reveal the extent and relative use of hierarchical action capacities in human and non-human animals.

## Methods
### Subjects
Human subjects were two expert flint knappers[16]. Chimpanzee subjects were 41 habituated wild western chimpanzees (*Pan trologyted verus*) (age 4–48 years), including both males and females from two communities (North and South) living in the Taï National Park, Ivory Coast[36]. Each day, T.P. conducted group follows from dawn until dusk and sampled nut-cracking behavior opportunistically during one nut-cracking season which lasted from January to April 2023.

### Procedure
**Chimpanzees.** 25.4 h of nut-cracking behavior was observed in total which contained 284 nutcracking action sequences which were live coded using a custom-built CyberTracker application on a smart-phone. A sequence began when an individual arrived at an anvil to crack a nut, and

**Table 1 | Definitions of each measurement or hierarchical complexity, the corresponding formula for calculating the measure, and a visual representation of a tree structure corresponding to a low and high value for each measurement**

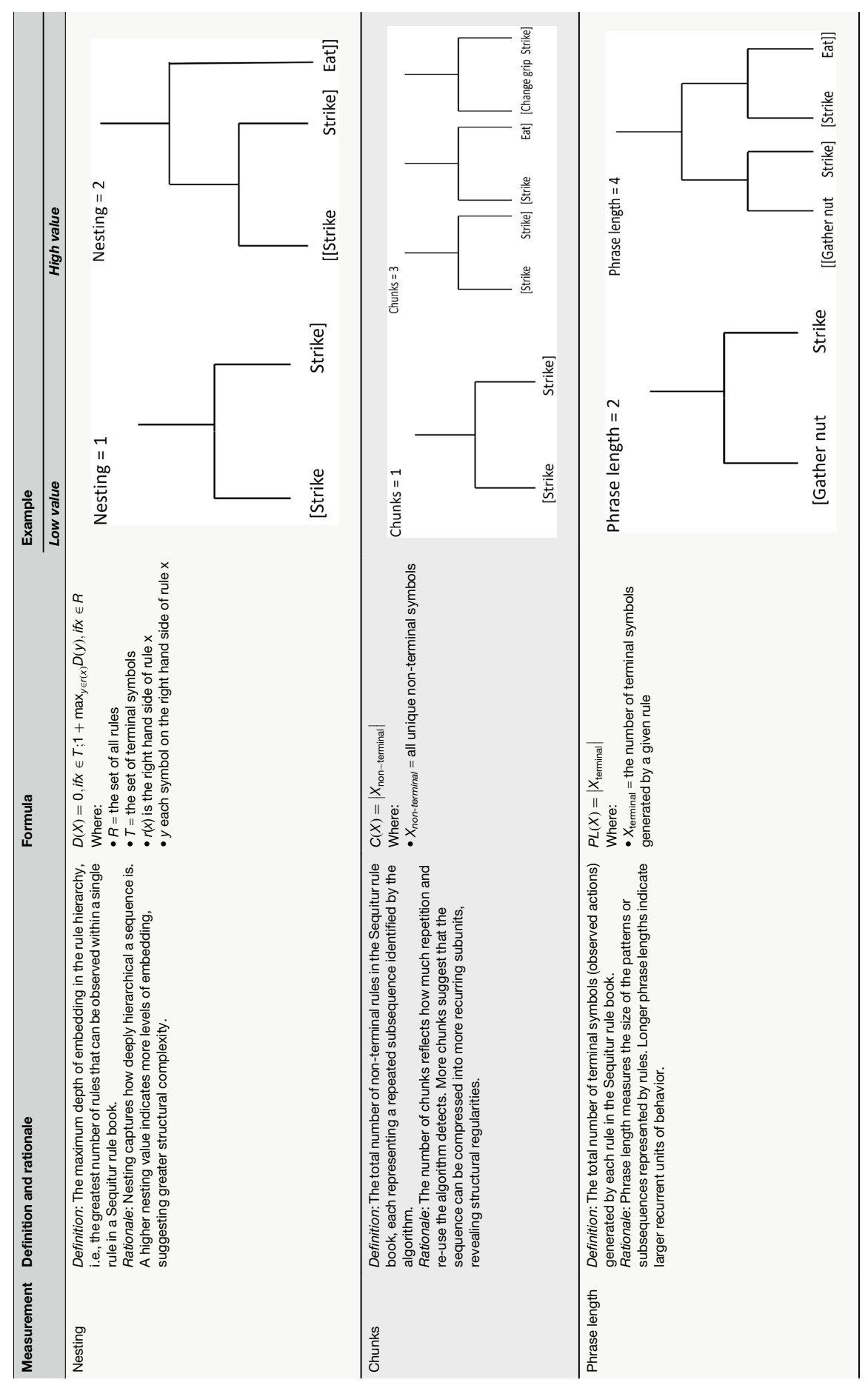

| Measurement | Definition and rationale | Formula | Example | |
|---|---|---|---|---|
| | | | *Low value* | *High value* |
| Nesting | *Definition*: The maximum depth of embedding in the rule hierarchy, i.e., the greatest number of rules that can be observed within a single rule in a Sequitur rule book. <br><br>*Rationale*: Nesting captures how deeply hierarchical a sequence is. A higher nesting value indicates more levels of embedding, suggesting greater structural complexity. | $D(X) = 0, \text{if } x \in T; 1 + \max_{y \in r(x)} D(y), \text{if } x \in R$ <br> Where: <br> • $R$ = the set of all rules <br> • $T$ = the set of terminal symbols <br> • $r(x)$ is the right hand side of rule x <br> • $y$ each symbol on the right hand side of rule x | | |
| Chunks | *Definition*: The total number of non-terminal rules in the Sequitur rule book, each representing a repeated subsequence identified by the algorithm. <br><br>*Rationale*: The number of chunks reflects how much repetition and re-use the algorithm detects. More chunks suggest that the sequence can be compressed into more recurring subunits, revealing structural regularities. | $C(X) = \lvert X_{non-terminal} \rvert$ <br> Where: <br> • $X_{non-terminal}$ = all unique non-terminal symbols | | |
| Phrase length | *Definition*: The total number of terminal symbols (observed actions) generated by each rule in the Sequitur rule book. <br><br>*Rationale*: Phrase length measures the size of the patterns or subsequences represented by rules. Longer phrase lengths indicate larger recurrent units of behavior. | $PL(X) = \lvert X_{terminal} \rvert$ <br> Where: <br> • $X_{terminal}$ = the number of terminal symbols generated by a given rule | | |

**Table 1 (continued) | Definitions of each measurement or hierarchical complexity, the corresponding formula for calculating the measure, and a visual representation of a tree structure corresponding to a low and high value for each measurement**

| Measurement | Definition and rationale | Formula | Example | |
|---|---|---|---|---|
| | | | Low value | High value |
| Structural entropy | *Definition*: The entropy of a binary encoding of the rule tree structure, where terminal nodes are coded as "1" and non-terminal nodes as "0." This quantifies the uncertainty or unpredictability of structural composition. *Rationale*: Entropy provides a measure of diversity and unpredictability in structural organization. Higher entropy suggests less predictable or more variable structure. | $H(X) = -\left(\frac{N_{count(0)}}{N}\log_2\frac{N_{count(0)}}{N} + \frac{N_{count(1)}}{N}\log_2\frac{N_{count(1)}}{N}\right)$ <br> Where: <br> • N = the total number of bits in the sequence <br> • Count(0) = the count of 0's in the sequence <br> • Count(1) = the count of 1's in the sequence | 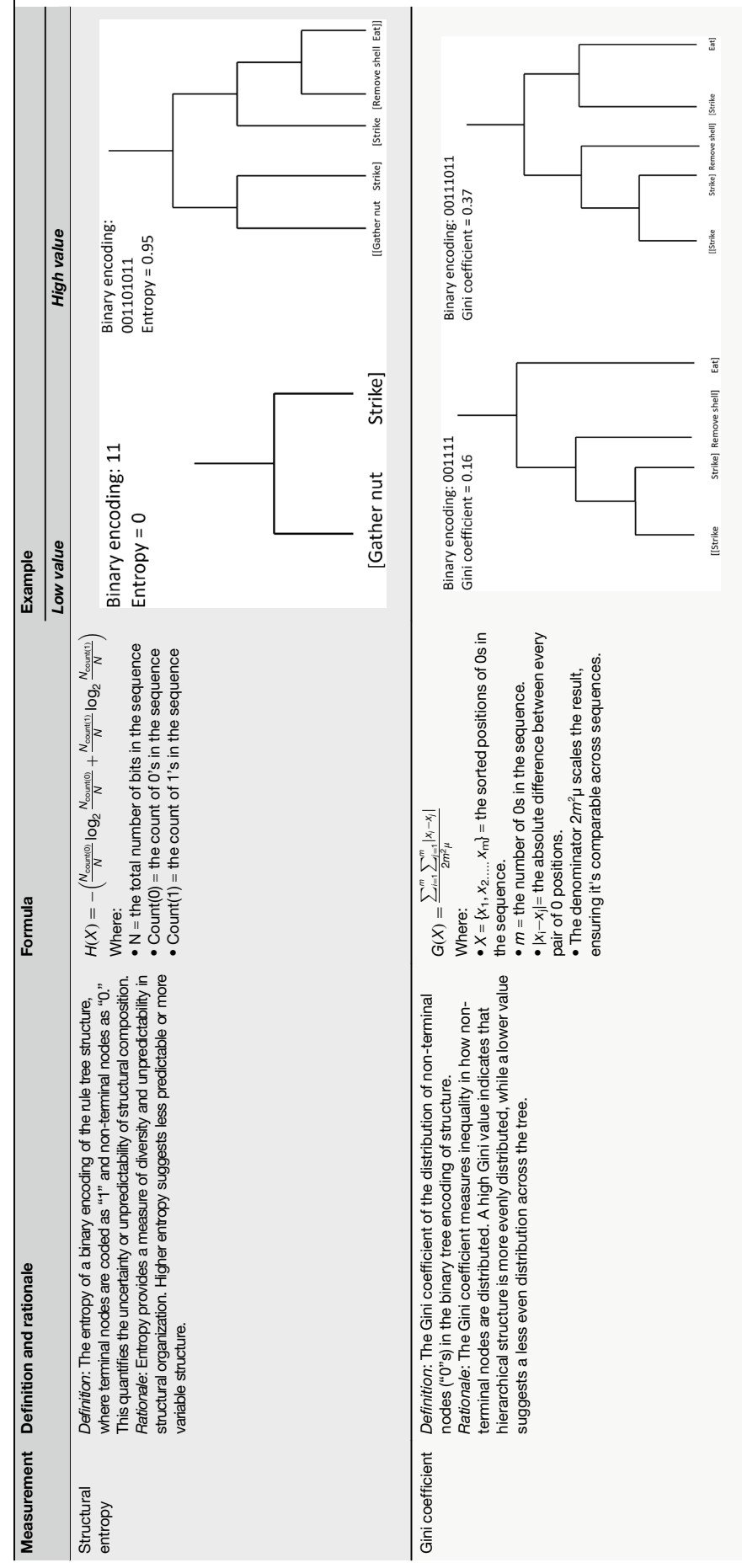 Binary encoding: 11 <br> Entropy = 0 | Binary encoding: 001101011 <br> Entropy = 0.95 |
| Gini coefficient | *Definition*: The Gini coefficient of the distribution of non-terminal nodes ("0"s) in the binary tree encoding of structure. *Rationale*: The Gini coefficient measures inequality in how non-terminal nodes are distributed. A high Gini value indicates that hierarchical structure is more evenly distributed, while a lower value suggests a less even distribution across the tree. | $G(X) = \frac{\sum_{i=1}^{m}\sum_{j=1}^{m}\|x_i - x_j\|}{2m^2\mu}$ <br> Where: <br> • $X = \{x_1, x_2, \ldots, x_m\}$ = the sorted positions of 0s in the sequence. <br> • m = the number of 0s in the sequence. <br> • $\|x_i - x_j\|$ = the absolute difference between every pair of 0 positions. <br> • The denominator $2m^2\mu$ scales the result, ensuring it's comparable across sequences. | Binary encoding: 001111 <br> Gini coefficient = 0.16 | Binary encoding: 00111011 <br> Gini coefficient = 0.37 |

**Table 1 (continued) | Definitions of each measurement or hierarchical complexity, the corresponding formula for calculating the measure, and a visual representation of a tree structure corresponding to a low and high value for each measurement**

| Measurement | Definition and rationale | Formula | Example | |
|---|---|---|---|---|
| | | | *Low value* | *High value* |
| Terminal : non-Terminal ratio | *Definition*: The ratio between terminal symbols (observed actions) and non-terminal symbols (rules) in the binary encoding of the rule tree. *Rationale*: This ratio provides a simple indicator of the balance between observed actions and the higher-level structures that generate them. A higher ratio indicates more non-terminal nodes relative to terminal symbols, suggesting deeper hierarchical organization | $R(X) = \frac{N(0)}{N(1)}$ Where : <br>• $N(0)$ = the number of 0's in the sequence. <br>• $N(1)$ = the number of 1's in the sequence. | Binary encoding: 0111 Node : Leaf ratio = 0.33 | Binary encoding: 01101011 Node : Leaf ratio = 0.6 |

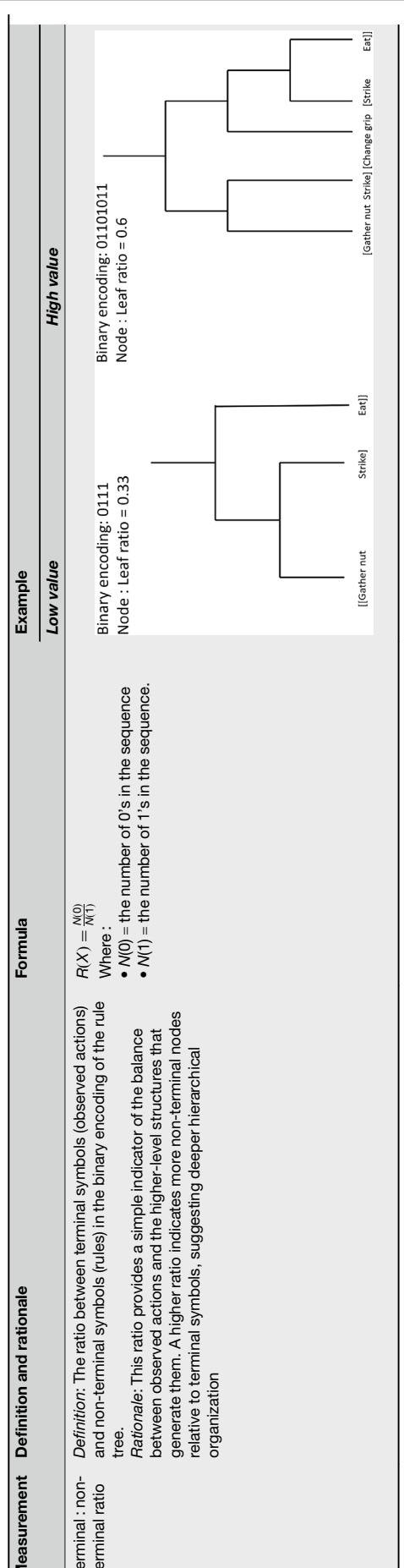

finished when the individual left the anvil, meaning multiple nuts could be cracked in a given sequence (watch nut cracking sequences on the Taï Chimpanzee Project channel: https://www.youtube.com/@taichimpanzeeproject7916). We had 1–15 sequences per individual ($M = 3.68$, SD = $\pm 3.44$). Seven behaviors were chosen to represent the sequence. The choice of behaviors was chosen to both accurately represent nut cracking sequence and also provide behaviors that closely correspond to behaviors coded in the human dataset. The ethogram is shown below and the corresponding behaviors in the human ethogram are written in brackets:

1) Gather nut: gathering an uncracked nut and placing it on an anvil.
2) Strike nut: striking nut with stone tool in hand (percussion).
3) Changing grip: changing grip on tool (grip shift tool).
4) Repositioning nut: repositioning the nut on the anvil (grip shift core).
5) Changing hand: changing the hand used to hold the tool (tool change).
6) Remove shell: removing the shell of the nut after cracking (remove flake).
7) Eat nut: eating nut after successfully cracking it.

**Humans.** Two expert flint-knappers were video recorded from a fixed position. Each participant manufactured nine oldowan and eight achulean stone-tools. The behavior of the participants was coded using an ethological approach within the software Etholog 2.25[37]. The following behaviors were coded:

1) Percussion: striking core with percussor (hammerstone or antler billet).
2) Target change: change in the location of percussion of the core.
3) Grip shift core: repositioning on the hand grasping the core.
4) Grip shift tool: repositioning of the hand grasping the percussor.
5) Inversion: flipping over the core without otherwise reorientating.
6) Flake detach: removal of a flake (judged to be) > 20 mm.
7) Tool change: exchange of one percussor for another.

For more details see original study[16].

**Simulating control sequences**

To establish whether chimpanzees structure their sequences hierarchically, it is necessary to compare not only to known hierarchically organized sequences (i.e., human action), but also to a known non-hierarchical baseline, since even random sequences will inevitably contain some degree of repetition by chance leading potentially to apparent hierarchical structure. As such, for each observed chimpanzee sequence, we simulated a randomly generated control sequence of exactly the same length. Since all behaviors are not equally probable to occur, we first calculated the probability of each behavior occurring in the overall set of observed sequences. Then, we randomly sampled from the resulting probability distribution of behaviors using a custom written r-script to generate control sequences exactly matching in length to each original sequence.

Random sequences lack structure, but linear sequences can also be non-random yet non-hierarchical. For example, Markov sequences are non-random, since sequences are emitted from a non-random probability distribution, yet they are not hierarchically structured since the current state in a Markov sequence depends only on the previous state. As such, we simulated Markov control sequences. We first trained a Markov model on chimpanzee sequences using the *Markovchain* package in R, which estimates the transition matrix by fitting the observed sequences to a Markov chain model. The trained model met the assumptions of Markov chains (i.e., it was both irreducible and aperiodic). From this model, we then generated a control sequence for each observed sequence of exactly the same length.

**Statistical analysis**

**Rule extraction.** Rules were extracted from linear action sequences using the lossless hierarchical compression algorithm 'Sequitur'[9]. The algorithm reads linear sequences of symbols from left to right. When a bigram of symbols is repeated, it is replaced by a rule that corresponds to that

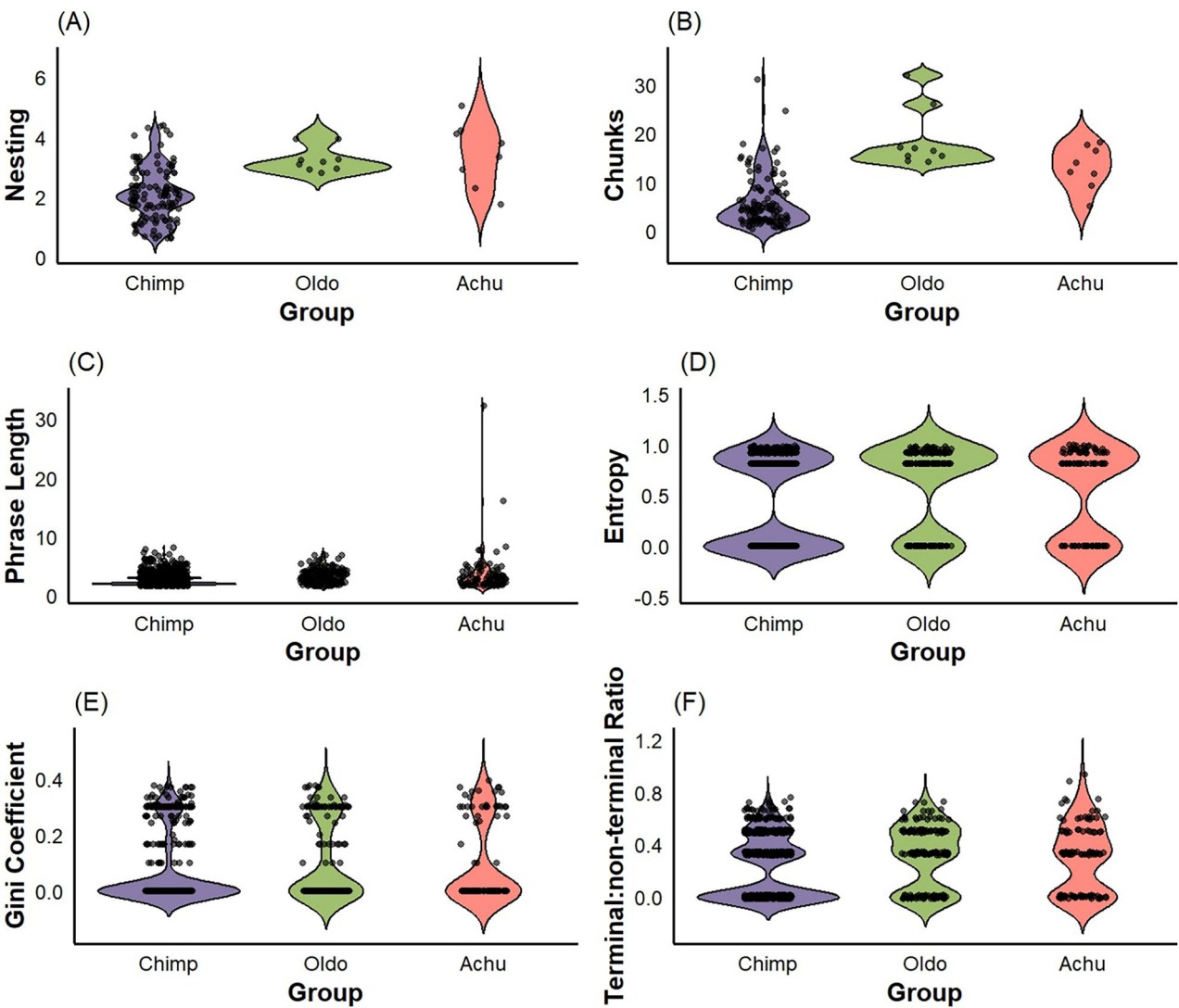

**Fig. 6 | Violin plots of hierarchical complexity of PTU action sequences in chimpanzees ($N = 140$) compared to Oldowan ($N = 9$) and Acheulean ($N = 8$) stone-tool manufacturing sequences.** Black dots represent original data points and the shape around them represents kernel density estimates. **A** Hierarchical depth of chimpanzee nut-cracking rules, (**B**) Number of rules per sequence, (**C**) Length of phrases generated by rules, (**D**) Entropy of tree structures, (**E**) Gini coefficient of tree structures, (**F**) Terminal:non-terminal ratio of tree structures.

bigram and the sequence is rewritten in a compressed format where a single symbol corresponding to a rule replaces the bigram. For example, the sequence "abdcab" would become "AdcA" where A corresponds to the repeating bigram "ab" from the original sequence. Through this process rules can become embedded within other rules. The number of times a bigram needs to be repeated to be considered a rule can be chosen by specifying a value for the parameter "K". In this paper we used $K = 2$, in line with a previous study of human stone-tool use[16]. The process is iteratively carried out until the sequence can no-longer be further compressed. The final output of the algorithm is the compressed sequence and the corresponding rule book. The analysis was run in MATLAB (v.R2024a) using a script found in the supplementary materials of Stout et al.[16].

**Measuring hierarchical complexity.** Hierarchical tree structures have many measurable dimensions. To comprehensively assess hierarchical complexity in sequences, we measure 6 dimensions of hierarchical complexity. We measured the number of rules found in each sequence (i.e., the number of "chunks"), the phrase length of each rule (i.e., the

length of the behavioral sequences each chunk corresponds to, which has a minimum of 2 since the sequitur algorithm builds rules from *pairs* of repeating behavior, but if one of those in the pair is itself a rule, phrase length may exceed 2) and the maximum number of nested dependencies within each rule-set (i.e., the rule with the largest number of rules embedded within it). These were our "basic" measures. We also took measures of rule structural complexity, by encoding each rule as a binary sequence, wherein 1 = terminal node and 0 = non-terminal node, and then measuring (1) the entropy of tree-structures, that is, the predictability of tree structures. (2) The "gini coefficient", which measures how (un)evenly distributed branches on the tree are, and (3) the ratio between terminal and non-terminal symbols in the tree structures (i.e., the balance of the internal structure of the tree, also known as "node:leaf" ratios). Definitions, formulas for calculation, and example tree structures are illustrated below in Table 1.

We included only sequences with a minimum of 8 actions because a sequence of fewer than 8 symbols cannot possibly generate a rule with a nested dependency within it[9]. This resulted in the removal of 144 sequences, leaving 140 for analysis. The number of rules, phrase length, and the

maximum number of nested dependencies were measured directly from Sequitur rule-books using a custom MATLAB script provided in "Supplementary Code 1". Structural entropy, gini coefficients, and terminal:non-terminal ratios were measured from binary encodings of tree structure which was extracted from sequitur rule-books, again using custom R-scripts that are provided in "Supplementary Code 1". The full processing pipeline is illustrated below in Fig. 6.

**Statistics and reproducibility.** We modeled our data with generalized linear models within a Bayesian framework. Three sets of comparisons were made: (1) between chimpanzee, Markov, and random sequences, (2) uniquely chimpanzee rule structures and those shared with Markov models, and (3) chimpanzee, Oldowan, and Acheulean PTU sequences. Models wherein chunks, nesting, and phrase-length were the dependent variables, a Poisson distribution with a log-link function was used. Models wherein entropy, gini coefficient, and terminal:non-terminal ratio were the dependent variable, a zero-inflated beta distribution with a log-link function was used. For all models, the intercept had a flat (uniform) prior, and the regression coefficient for the categorical predictor had a weakly informative normal (0,10) prior. Models were run using 4 chains, 2000 iterations, and an additional 1000 warm-up iterations. Model fit was evaluated using Rhat values, effective sample size estimates, p-p plots, and trace-rank plots, which showed all models were a good fit to the data. All models were run using the brms package[38] in RStudio (v.2023.03) and results are presented with a credible interval of 95%. The code for running the models can be found in "Supplementary Code 2", the code for reproducing the figures can be found in "Supplementary Code 3" and the data can be found in "Supplementary Data 1".

**Ethics statement.** We have complied with all relevant ethical regulations for animal use. The Tai Chimpanzee Project is committed to non-invasive research, not harming research animals. Data collection for this study comes from long-term data and/or other completely observational focal follows. The research methods have been approved by the *Ethikrat der Max Planck Gesellschaft* (Ethikrat-MPG-04082014). Focal follows are conducted under strict hygiene rules, adopted by the *IUCN Best Practice Guideline for Health Monitoring and Disease Control in Great Ape Populations (2015)*, including a 5-day quarantine, keeping seven meters distance, and wearing a surgical face mask.

For the human sample, all ethical regulations relevant to human research participants were followed and informed consent was obtained. Ethical approval was gained from the University College London Research Ethics Committee [0603/001][16].

### Reporting summary
Further information on research design is available in the Nature Portfolio Reporting Summary linked to this article.

## Data availability
All data used in this study are freely available in the supplementary materials and can also be found on figshare: https://doi.org/10.6084/m9.figshare.30862226. All data is source data.

## Code availability
All code used in this study is freely available in the supplementary materials and can also be found on figshare: https://doi.org/10.6084/m9.figshare.30862226. All code is source code.

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

## Author contributions

D.T. designed the study, did the analysis, and wrote the paper. T.P. designed the study, collected the data, and edited the manuscript. C.C. designed the study and edited the manuscript. R.W. designed the study and edited the manuscript.

## Funding

## Competing interests

The author declares no competing interests.
