## [Transparent Peer Review file · Communications Biology]

MEASURING HIERARCHICAL STRUCTURE IN SEQUENTIAL BEHAVIOUR: HOMINOID PERCUSSIVE TOOL-USE COMPARED

Corresponding Author: Dr Derry Taylor

Version 0:

Reviewer comments:

Reviewer #1

(Remarks to the Author)

1. Summary

The MS outlines a new analysis pipeline for identifying hierarchical structure in behaviour sequences. It then offers a two-part proof of concept for the analysis. First, the hierarchical complexity in real chimpanzee sequences was compared with results produced by a Markov model trained on chimpanzee percussive tool-use data. The latter represents a control, as Markov processes do not produce hierarchical structuring. Next the real chimpanzee sequences are compared with Oldowan and Acheulean percussive techniques, which are known to have hierarchical structure. Results show that chimpanzee percussive tool-use displays lower hierarchical complexity than that of Early Stone Age percussive techniques.

2. Impression

For the most part, I think the MS is well-conceived, well-written and potentially represents an important new tool in analysing hierarchical structure in behaviour sequences. From the detail provided, it should reproduce the work and apply it to different animals/behaviours. However, I think more needs to be said regarding the novelty and implications of the paper.

3. Comments

Major

#1: In the abstract, the MS states "It has long been theorized that cognitive capacities are revealed through the sequential structure of natural behavior, particularly its hierarchical organization. Progress in understanding the origins of this capacity has, however, been limited by a lack of techniques for identifying and measuring hierarchical structure in behavioural sequences." It goes on to state that the new methodology might potentially "play a pivotal role in shedding new light on old questions." I think the impact of the paper would be greater if the MS spelled out in more detail precisely the link between hierarchy in behavioural sequences and cognitive capacities. The discussion section is short, and mainly just notes the link between hierarchically structured behaviour and Broca's area. Here is one suggestion (though there are many ways of responding to this comment). One issue (which I suspect will interest many) is the link between hierarchically structured behaviour and hierarchical cognition. Does the new analysis pipeline suggest that chimpanzee nut-cracking is the product of hierarchically organised representations, rather than associative chaining? If the MS was to engage with this debate, it might be productive to look at Enquist, M. et al (2023).

#2, pge 2: Stout (2011) quantifies the complexity of action hierarchies for the Oldowan and Acheulean, and then uses it to argue against claims that the lower palaeolithic was technologically static. I think the MS should engage with this work to some degree at least, if only to illustrate the novelty/benefits of the new methodology proposed.

Minor

#1, pge 2: "Similarly, human tool-use sequences have been claimed to be hierarchically organized (Planer & Sterelny, 2021)" In conjunction with the first sentence of the paper (regarding human uniqueness), this makes it look like Planer and Sterelny argue the cognitive capacities underlying syntax production are human-unique. But in fact they argue that these are (quantitatively) elaborated capacities we share with the great apes. To put this another way: while language may be human-unique, the point of most tool-language co-evolutionary hypotheses is to show that the hierarchical cognition found in tool-making and -use can be elaborated via selection to capacities for syntax production. In this sense, those capacities are often

thought not to be human-unique. I think the MS could be clearer on these issues.

#2, pge 2: "However, owing to a lack of techniques for identifying and measuring hierarchical structure in natural sequences of behaviour, this claim has continued to be contested (Moro, 2014)." I don't see where Moro makes this claim in his paper; rather, his argument is that language and action are too disanalogous for their to be any evolutionary relationship between them. Berwick and Chomsky, in their 2016 book, make similar points. The MS could be clearer here about the need for the new analysis pipeline.

References

Enquist, M., Ghirlanda, S., & Lind, J. (2023). *The Human Evolutionary Transition: From Animal Intelligence to Culture*. Princeton University Press.

Stout, D. (2011). Stone toolmaking and the evolution of human culture and cognition. *Philosophical Transactions: Biological Sciences*, 366(1567), 1050–1059.

Reviewer #2

(Remarks to the Author)

In the manuscript "MEASURING HIERARCHICAL STRUCTURE IN SEQUENTIAL BEHAVIOUR: HOMINOID PERCUSSIVE TOOL-USE COMPARED", authors make three important claims:

- First, that they develop a new methodological framework that enables quantitative analysis of behavioural hierarchies;
- Second, that chimpanzee percussive tool-use (PTU) sequences exhibit hierarchical structuring;
- Third, that chimpanzee hierarchical complexity is systematically lower than humans.

The type of approach used, in particular the observation without intervention of chimpanzees in conservation areas is interesting and deserve full attention, if only considering the difficulty of running these types of ethological observations. In addition, the observation of simple action structure in chimpanzees use of stones to crack nuts open really deserves a serious consideration.

As a reviewer, I have serious concerns regarding the other points, that in my opinion impede the publication of this current manuscript.

Regarding the first point, the very strong claim "we introduce here a new analysis pipeline" contradicts the fact that one already published manuscript, the Stout et al., 2021, already introduced hidden Markov Models to investigate the structure of complex tool-use sequences of actions. Compared to this previous highly relevant and I suppose inspiring work (it is cited throughout the manuscript) it isn't clear how the current manuscript really brings something new.

Regarding the third point, there are reasons to debate whether the chimpanzee's nut-cracking and humans stone-tool making can be compared in the way it is proposed in the manuscript. There are at least two issues with the comparisons as it is presented.

First, the equivalences between human and chimpanzees ethogram are not sufficiently defended, as there as many different behaviours that are considered as equivalent between the two species, almost all of actions 3 to 6 could be argued not to be equivalent between the human and behaviour behaviours, making comparison of the hierarchical structure of the behaviours difficult to argue.

Second, the authors mix Acheulean and Oldowan stone-tool making as human behaviours for comparisons to chimpanzees' nut-cracking behaviours, but these are clearly very different behaviours, not only in terms of the final product but also iin terms of evolutionary history, Oldwoan stone-tools preceding Acheulean stone-tools by ages f human evolutions. Hence, the results could have been revealing if chimpanzees but-cracking and human Oldowan stone-tool making, could have been directly compared as both share close behavioural features such as percussion for removal of parts of a target object (nuts for chimpanzees, the core for Oldowan stone-tool making).

The choice instead of presenting the two technologies as "cases" of humans stone tools, especially in figure 4 were Acheulean on the left and Oldowan on the right flank the chimpanzees behaviour, plays against such direct comparison. There are also some minor typographic errors, such as a carriage return missing in p5, l13.

Reviewer #3

(Remarks to the Author)

This paper introduces a method for detecting and quantifying hierarchical structure in behavioral sequences. It is applied to chimpanzee nut-cracking and human stone tool-making behaviors. The authors argue that chimpanzee action sequences exhibit a level of hierarchical complexity that surpasses what could be generated by random processes or simple Markov models. They further claim that human sequences display even greater hierarchical complexity than those of chimpanzees.

Overall, I find the study well-motivated and the research question important. However, I have some significant concerns regarding aspects of the statistical analyses. Unless the authors are using methods that I am unfamiliar with and can justify, I believe the statistical component of the study may require substantial revision.

Here are my concerns regarding the analyses:

1) The authors generate sequences using random and Markov processes to test whether the chimpanzee and human sequences exhibit greater hierarchical complexity than would be expected by chance, because random and Markov processes should not produce, except by chance, hierarchical structure. However, they then include these generated sequences in the main dataset and fit linear models to assess the effect of sequence origin (real vs. random simulated vs markov simulated) on measures of hierarchical complexity. This raises a concern: the alpha values and confidence intervals derived from the regression may be sensitive to the number of simulated sequences included, potentially introducing arbitrariness into the results. Can the authors clarify whether—and how—the number of simulated sequences affects the estimated alpha values and confidence intervals?

An alternative approach that might better capture the logic of the comparison would be to use the random or the Markov-generated sequences to build null distributions for each hierarchical structure measure. By simulating a large number of sequences under each generative process, the authors could estimate the expected distribution of hierarchical structure measures under the null hypothesis. The observed hierarchical structure measures for the real chimpanzee and human data could then be compared to these null distributions — for example, by computing p-values or percentile ranks — to formally assess whether the real sequences exhibit significantly greater hierarchical structure.

2) The authors compare human and chimpanzee sequences, but the tasks performed by each species differ, making it unclear what the observed differences in hierarchical complexity truly reflect. While the authors argue that chimpanzee actions correspond to human actions, the underlying tasks are not totally equivalent. Do the differences observed indicate a disparity in the capacity for hierarchical structuring between species, or do they simply reflect the distinct hierarchical demands of the respective tasks?

3) When the authors generate random sequences for comparison, they draw integers from a uniform distribution. However, this choice may be problematic: ideally, the probabilities of drawing each integer in the simulated sequences should reflect the empirical frequencies observed in the chimpanzee nut-cracking and human stone-tool-making datasets.

4) It seems that the algorithm identifies exact repetitions of action sequences, such as [place nut][strike nut][eat nuts]. However, depending on task conditions, functionally equivalent behaviors may differ slightly in their execution. For instance, if a nut is unusually hard or a strike is missed, the sequence might appear as [place nut][strike nut][strike nut][eat nuts]. While this variation preserves the underlying structure, the algorithm would treat it as a distinct sequence due to the additional action, potentially missing structural similarities. This potential limitation does not appear to be directly addressed in the current manuscript.

Additional comments:

- Briefly clarify what the 'markovchain' function does. I assume it fits the data to estimate the transition matrix, but it would be helpful for a broader audience if this were stated explicitly.

- The meaning of "the phrase length of each rule" is unclear for me — I had assumed that each rule was of length 2. Please clarify what is meant by this expression.

- The definitions of the measures of complexity are difficult to understand. It would be helpful if the authors could provide a clearer explanation of how these measures are defined and the rationale behind their definition.

- The section "Analyzing hierarchical complexity" would benefit from greater clarity and precision. For example, the authors could provide more explicit definitions of the dependent and explanatory variables used in their models and clarify the interpretation of the alpha value.

Typos:

Remove "our" (p.3, l.7)

Add "7)" before "Tool change" and move to a new line (p.5, l.13)

Version 1:

Reviewer comments:

Reviewer #1

(Remarks to the Author)

I think the authors have done a good job of responding to my concerns, and that the MS as a whole reads much better.

Reviewer #2

(Remarks to the Author)

The authors fully addressed the original comments and the manuscript should be accepted for publication in my opinion.

Reviewer #3

(Remarks to the Author)

I have read the authors' response to my comment and am satisfied with their reply.

Measuring hierarchical structure in sequential behaviour: hominoid percussive tool-use compared

We would like to thank all the Reviewers and the Editor for their useful comments, which greatly helped us to improve our previous manuscript. Below, we discuss the Reviewers comments in detail. For convenience, we reproduce the Reviewers' comments in italics (marked by "Reviewer comment x:"), followed by our response (marked by "Author response:"). We numbered Reviewers' comments continuously in order to facilitate referring back to previous Reviewer comments and Author responses (points), as in the case when Reviewer 2 or Reviewer 3 made a comment similar to a comment made by Reviewer 1 or vice versa.

Reviewer #1

Reviewer comment 1: *The MS outlines a new analysis pipeline for identifying hierarchical structure in behaviour sequences. It then offers a two-part proof of concept for the analysis. First, the hierarchical complexity in real chimpanzee sequences was compared with results produced by a Markov model trained on chimpanzee percussive tool-use data. The latter represents a control, as Markov processes do not produce hierarchical structuring. Next the real chimpanzee sequences are compared with Oldowan and Acheulean percussive techniques, which are known to have hierarchical structure. Results show that chimpanzee percussive tool-use displays lower hierarchical complexity than that of Early Stone Age percussive techniques.*

Author response: We thank reviewer 1 for the accurate summary of our paper.

Reviewer comment 2: *For the most part, I think the MS is well-conceived, well-written and potentially represents an important new tool in analysing hierarchical structure in behaviour sequences. From the detail provided, it should reproduce the work and apply it to different animals/behaviours. However, I think more needs to be said regarding the novelty and implications of the paper.*

Author response: We thank the reviewer for their positive assessment of the manuscript and for highlighting the need to expand on the novelty and implications of our work. In response, we have substantially strengthened the discussion to emphasize the broader significance of the approach. In particular, we now highlight that the key contribution of our pipeline lies in providing *domain-general, cross-comparable measures* of hierarchical structure and complexity in sequential behaviour. By focusing on structural properties rather than task-specific content, the method enables meaningful comparisons across different behaviors, species, and modalities. We argue that this generality is novel because it bridges empirical behavioral analysis with larger theoretical debates about the evolution of language, the cognitive foundations of cultural innovation, and the principles governing complex sequence organization across biological and cultural systems (p29, line 12 – p30, line 6). We believe that this addition directly addresses the reviewer's concern by clarifying both the distinctive contribution of the method and its broader implications for comparative and evolutionary research.

Reviewer comment 3: *In the abstract, the MS states "It has long been theorized that cognitive capacities are revealed through the sequential structure of natural behavior, particularly its hierarchical organization. Progress in understanding the origins of this capacity has, however, been limited by a lack of techniques for identifying and measuring hierarchical structure in behavioural sequences." It goes on to state that the new methodology might potentially "play a pivotal role in shedding new light on old questions." I think the impact of the paper would be greater if the MS spelled out in more detail precisely the link between hierarchy in behavioural sequences and cognitive*

capacities. The discussion section is short, and mainly just notes the link between hierarchically structured behaviour and Broca's area. Here is one suggestion (though there are many ways of responding this comment). One issue (which I suspect will interest many) is the link between hierarchically structured behaviour and hierarchical cognition. Does the new analysis pipeline suggest that chimpanzee nut-cracking is the product of hierarchically organised representations, rather than associative chaining? If the MS was to engage with this debate, it might be productive to look at Enquist, M. et al (2023).

#2, page 2: Stout (2011) quantifies the complexity of action hierarchies for the Oldowan and Acheulean, and then uses it argue against claims that the lower palaeolithis was technologically static. I think the MS should engage with this work to some degree at least, if only to illustrate the novelty/benefits of the new methodology proposed.

Author response: We thank the reviewer for this valuable suggestion, which has significantly strengthened the discussion. We have substantially revised the section to explicitly engage with debates about whether non-human primate action sequences reflect associative chaining or hierarchically organized representations. The revised discussion now:

1. **Directly addresses Enquist et al:** We contextualize their simulations of nut-cracking, highlighting that while their work shows chaining and conditioned reinforcement can explain highly simplified cases, these simulations assume a single fixed solution. In contrast, natural nut-cracking is ecologically variable (e.g., differing anvil types, tool orderings), which our data show gives rise to diverse and complex action sequences. We explain how this ecological variability and flexibility likely impose planning demands beyond simple chaining, helping to account for why our Markov models captured much of the observed structure but failed to model the more hierarchically complex rules.
2. **Engages with Stout (2011):** We explicitly compare our fully automated, scalable approach to Stout's archaeological analyses, noting that our pipeline complements and extends this foundational work by inferring hierarchical structure directly from behavioral data and enabling quantitative model-based comparisons across species.
3. **Clarifies cognitive implications and limits:** We added explicit statements about what can and cannot be concluded from our results. We emphasize that our findings provide strong evidence that chimpanzee sequences exhibit hierarchical organization beyond local associations, but that we do not claim direct evidence of tree-like mental representations, acknowledging that similar behavioral structure could arise from chunking, procedural routines, or biomechanical constraints.
4. **Expands cognitive framing throughout the discussion:** The section now more clearly links our findings to debates on hierarchical cognition, tool-use planning, and evolutionary differences in cognitive architecture, while still emphasizing the methodological contribution of our approach.

We believe these changes (p27 line 6 – p28 line 8) directly address the reviewer's concerns by making the manuscript's cognitive contributions clearer, engaging with key literature, and situating the novelty of our pipeline in the context of existing debates.

Reviewer comment 4: page 2: "Similarly, human tool-use sequences have been claimed to be hierarchically organized (Planer & Sterelny, 2021)" In conjunction with the first sentence of the paper (regarding human uniqueness), this makes it look like Planer and Sterelny argue the cognitive capacities underlying syntax production are human-unique. But in fact they argue that these are (quantitatively) elaborated capacities we share with the great apes. To put this another way: while

language may be human-unique, the point of most tool-language co-evolutionary hypotheses is to show that the hierarchical cognition found in tool-making and -use can be elaborated via selection to capacities for syntax production. In this sense, those capacities are often thought not to be human-unique. I think the MS could be clearer on these issues.

Author response: We thank the reviewer for pointing this out and have revised the introduction to better reflect Planer and Sterelny's position and the broader consensus in the literature. The new text clarifies that while language itself is human-unique, the underlying hierarchical action planning capacities are likely shared with other great apes and have been quantitatively elaborated in humans. We now explicitly state that tool-use sequences provide a window into these shared capacities, which supports both co-evolutionary and homologous hypotheses rather than implying a sharp human/non-human divide. This clarification ensures our framing aligns with existing literature and avoids misrepresenting Planer and Sterelny's argument (p2 line 2 – p2 line 16).

Reviewer comment 5: *pge 2: "However, owing to a lack of techniques for identifying and measuring hierarchical structure in natural sequences of behaviour, this claim has continued to be contested (Moro, 2014)." I don't see where Moro makes this claim in his paper; rather, his argument is that language and action are too disanalogous for their to be any evolutionary relationship between them. Berwick and Chomsky, in their 2016 book, make similar points. The MS could be clearer here about the need for the new analysis pipeline.*

Author response: We thank the reviewer for pointing out our unclear statement of the view purported in Moro (2014). This is, in fact, a miscommunication. The sentence makes two points, firstly, that there is a lack of techniques for identifying and measuring hierarchical structure in sequential behaviour, and second, that the claim that language and action share similarity in structure continues to be contested. Our intention was to cite Moro (2014) as an example of the ongoing debate, not to suggest that Moro (2014) claims the reason is due to a lack of techniques. As the reviewer points out, Moro's (2014) claim is more that these systems are fundamentally different from a more conceptual/theoretical point of view. So, we have broken this sentence down into two parts, one suggesting that there is a lack of standardized techniques for identifying and measuring hierarchical structure, and a second, suggesting that it is therefore not surprising that the debate on the relationship between language and action continues (p2 line 13 – p2 line 16). We hope this resolves the issue.

Reviewer #2

Reviewer comment 6: *In the manuscript "MEASURING HIERARCHICAL STRUCTURE IN SEQUENTIAL BEHAVIOUR: HOMINOID PERCUSSIVE TOOL-USE COMPARED", authors make three important claims:*

- First, that they develop a new methodological framework that enables quantitative analysis of behavioural hierarchies;*
- Second, that chimpanzee percussive tool-use (PTU) sequences exhibit hierarchical structuring;*
- Third, that chimpanzee hierarchical complexity is systematically lower than humans.*

The type of approach used, in particular the observation without intervention of chimpanzees in conservation areas is interesting and deserve full attention, if only considering the difficulty of running these types of ethological observations. In addition, the observation of simple action structure in chimpanzees use of stones to crack nuts open really deserves a serious consideration.

Author response: We thank the reviewer for their positive assessment of our manuscript. Regarding the point of simple action structure, we have included a new discussion paragraph which directly addresses this issue. We discuss previous simulations of chimpanzee nut-cracking that show this behaviour can be learned through conditioned reinforcement and chaining (Enquist et al., 2016). While our finding that around 70% of the rules used by chimpanzees are captured by a Markov model indicates a relatively simple structure, we note that ecologically valid settings introduce variability in tool and substrate affordances, which require more complex sequential organization of behaviour. Accordingly, our analysis demonstrates that chimpanzee nut-cracking exhibits hierarchical structure beyond what the Markov model can capture (p27 line 6 – p28 line 8).

We also clarify that the aim of the paper is *not* to compare chimpanzee hierarchical organization to humans per se, but to show that chimpanzees exhibit measurable hierarchical organization. Our method allows for quantifying the depth and complexity of sequences, enabling comparisons across different tool-use behaviours within a species, and potentially across taxa and tool-use contexts (e.g., chimpanzee nut-cracking versus hominin percussive tool-making). In this way, the study demonstrates the broader applicability of our pipeline for assessing hierarchical complexity in sequential behaviour, without overinterpreting interspecies differences. We have modified our discussion to communicate this (p26 line 12 – p28 line 5; p28 line 19 – p28 line 22).

Reviewer comment 7: *As a reviewer, I have serious concerns regarding the other points, that in my opinion impede the publication of this current manuscript.*

Regarding the first point, the very strong claim "we introduce here a new analysis pipeline" contradicts the fact that one already published manuscript, the Stout et al., 2021, already introduced hidden Markov Models to investigate the structure of complex tool-use sequences of actions. Compared to this previous highly relevant and I suppose inspiring work (it is cited throughout the manuscript) it isn't clear how the current manuscript really brings something new.

Author response: We thank the reviewer for drawing our attention to the need to be much clearer about what it is that is new here. In brief, previous approaches, such as Stout and colleagues (2021) looked at compressibility (i.e., how much shorter can we make a sequence by replacing repetitions with a rule that corresponds to the repetitions). However, this is an indirect and in some cases misleading measurement of hierarchical structure. As illustrated in a new figure 1 which we have added to the introduction, there are cases where you can have two sequences of equal length, which are equally compressible, but one contains hierarchical structure (that is, rules embedded within other rules), while the other does not. Moreover, there are many possible ways in which the same sequence could be organized hierarchically, which differ with respect to hierarchical depth and patterning (illustrated in the new figure 2), which carry different implications for the underlying computation and cognition. So, we need direct measurements of hierarchical structure, and that is what we provide here – a way to take a diverse range of measurements of hierarchical structure and complexity, which are validated by our demonstration that according to these measurements chimpanzee nut cracking contains a hierarchical complexity beyond what can be produced by a markov chain which cannot theoretically produce hierarchical structuring as these models are memory-less and hierarchical structuring involves dependencies over long sequential distances. This is explained from p3 line 1 – p4 line 1.

Reviewer comment 8: *Regarding the third point, there are reasons to debate whether the chimpanzee's nut-cracking and humans stone-tool making can be compared in the way it is proposed in the manuscript. There are at least two issues with the comparisons as it is presented.*

First, the equivalences between human and chimpanzees ethogram are not sufficiently defended, as there as many different behaviours that are considered as equivalent between the two species,

almost all of actions 3 to 6 could be argued not to be equivalent between the human and behaviour behaviours, making comparison of the hierarchical structure of the behaviours difficult to argue. Second, the authors mix Acheulean and Oldowan stone-tool making as human behaviours for comparisons to chimpanzees' nut-cracking behaviours, but these are clearly very different behaviours, not only in terms of the final product but also in terms of evolutionary history, Oldowan stone-tools preceding Acheulean stone-tools by ages of human evolutions. Hence, the results could have been revealing if chimpanzees nut-cracking and human Oldowan stone-tool making, could have been directly compared as both share close behavioural features such as percussion for removal of parts of a target object (nuts for chimpanzees, the core for Oldowan stone-tool making). The choice instead of presenting the two technologies as "cases" of human stone tools, especially in figure 4 where Acheulean is on the left and Oldowan on the right flank the chimpanzee behaviour, plays against such direct comparison.

Author response: We appreciate this thoughtful comment and have made several changes in response.

First point: We have strengthened the rationale for our behavioural comparisons by citing de la Torre & Hirata (2015), who argue that hominin stone tool manufacture represents a direct elaboration of the percussive abilities of ancestral primates. In addition, we have revised the Methods section to clarify that the chimpanzee behaviours are presented as *closely related* rather than *directly equivalent* to the human behaviours.

Second point: We have substantially revised the "Hominoid PTU compared" section of the Results. In the revised text, we explicitly highlight comparisons not only between chimpanzee nut-cracking and human stone tool use but also between chimpanzee nut-cracking and Oldowan manufacture, as well as between Oldowan and Acheulean tool-making. This expanded comparison reveals a noteworthy pattern: Oldowan behaviours often occupy an intermediate position in sequential complexity between chimpanzee nut-cracking and Acheulean biface production, supporting the view that Oldowan technology represents an elaboration of earlier percussive capacities. We also explicitly discuss this interpretation in the revised Discussion section (page 26 line 2 – page 27 line 5), although we caution that this is not our explicit aim, our aim is to illustrate our method shows the type of comparative differences we should expect to see, demonstrating the scope for comparative studies across tasks and species, meaning we cannot claim systematic species differences on this basis.

We appreciate the suggestion to juxtapose chimpanzee nut-cracking directly with Oldowan technology in Figure 4 (now figure 6) and have reorganized the figure accordingly.

Reviewer comment 9: *There are also some minor typographic errors, such as a carriage return missing in p5, l13.*

Author response: Thank you for pointing this out, it has now been corrected.

Reviewer #3

Reviewer comment 10: *This paper introduces a method for detecting and quantifying hierarchical structure in behavioral sequences. It is applied to chimpanzee nut-cracking and human stone tool-making behaviors. The authors argue that chimpanzee action sequences exhibit a level of hierarchical*

complexity that surpasses what could be generated by random processes or simple Markov models. They further claim that human sequences display even greater hierarchical complexity than those of chimpanzees.

Overall, I find the study well-motivated and the research question important. However, I have some significant concerns regarding aspects of the statistical analyses. Unless the authors are using methods that I am unfamiliar with and can justify, I believe the statistical component of the study may require substantial revision.

Author response: We very much appreciate the positive assessment of the study topic and we believe we have fully addressed the reviewers statistical concerns in the revisions made in response to the comments below.

Reviewer comment 11: *The authors generate sequences using random and Markov processes to test whether the chimpanzee and human sequences exhibit greater hierarchical complexity than would be expected by chance, because random and Markov processes should not produce, except by chance, hierarchical structure. However, they then include these generated sequences in the main dataset and fit linear models to assess the effect of sequence origin (real vs. random simulated vs markov simulated) on measures of hierarchical complexity. This raises a concern: the alpha values and confidence intervals derived from the regression may be sensitive to the number of simulated sequences included, potentially introducing arbitrariness into the results. Can the authors clarify whether—and how—the number of simulated sequences affects the estimated alpha values and confidence intervals?*

An alternative approach that might better capture the logic of the comparison would be to use the random or the Markov-generated sequences to build null distributions for each hierarchical structure measure. By simulating a large number of sequences under each generative process, the authors could estimate the expected distribution of hierarchical structure measures under the null hypothesis. The observed hierarchical structure measures for the real chimpanzee and human data could then be compared to these null distributions — for example, by computing p-values or percentile ranks — to formally assess whether the real sequences exhibit significantly greater hierarchical structure.

Author response: We appreciate this suggestion and understand the concern that regression estimates can, in principle, be sensitive to sample size. In our analysis, this is unlikely to introduce bias for two reasons. First, we used a matched control design: for each observed sequence, we generated one random and one Markov sequence of identical length, ensuring equal numbers of sequences across conditions and eliminating confounds due to unequal sample sizes. Second, all models converged well and provided accurate predictions, indicating that the sample size was sufficient for stable parameter estimates. Because of this matched design and strong model performance, we are confident that our alpha values and confidence intervals are robust.

We also agree that building null distributions from random or Markov-generated sequences, and then comparing observed hierarchical measures to these null distributions (e.g., via percentile ranks or p-values), is a valid alternative approach. However, such null-distribution approaches typically pool sequences of varying lengths and other properties, and therefore do not preserve the pairwise relationship between a given real sequence and a matched simulated sequence. In contrast, our matched-design regression directly compares each real sequence to a random and Markov sequence

of the same length, controlling for sequence-level variability and length. This enables interpretable effect sizes and confidence intervals while preserving the natural variability structure of the observed sequences. Additionally, our approach is computationally more efficient, as it does not require simulating large numbers of sequences to stabilize percentile estimates, while still capturing the same logic of “real vs. chance” comparisons.

While implementing null-distribution testing would be feasible and could provide complementary information, we believe our matched-design regression provides a rigorous, balanced, and efficient assessment of hierarchical complexity, with the added benefit of directly interpretable effect sizes and confidence intervals.

Reviewer comment 12: *The authors compare human and chimpanzee sequences, but the tasks performed by each species differ, making it unclear what the observed differences in hierarchical complexity truly reflect. While the authors argue that chimpanzee actions correspond to human actions, the underlying tasks are not totally equivalent. Do the differences observed indicate a disparity in the capacity for hierarchical structuring between species, or do they simply reflect the distinct hierarchical demands of the respective tasks?*

Author response: We agree that the tasks compared here—chimpanzee nut-cracking and human Oldowan and Acheulean tool manufacture—are not perfectly equivalent, and this is an inherent limitation of cross-species comparisons. Our aim is not to make claims about absolute species-level capacities, but rather to examine hierarchical organization in representative percussive tool-use behaviors. In this sense, the comparison focuses on whether the hierarchical complexity of specific tool-use tasks differs across taxa, rather than inferring the overall cognitive limits of chimpanzees or humans.

Percussive technology provides a meaningful comparative framework because it occurs across hominoids and underpins lithic manufacture, allowing us to investigate the hierarchical structure of tool-use behavior in both chimpanzees and early humans in a functionally analogous context. Experimental work with captive great apes (e.g., Toth et al., 1993) shows that they can acquire basic stone flaking skills under intensive training, indicating that task-specific capacities may extend beyond what is observed in the wild, but our study does not attempt to quantify those potential limits.

We have clarified this framing in the Discussion (p26 line 1 – p27 line 5; p28 line 19 – p28 line 22), explicitly noting that the observed differences in hierarchical complexity likely reflect both the specific hierarchical demands of the tasks and the species performing them, rather than absolute species-level abilities. This task-focused, cross-taxa comparison highlights how percussive tool-use behavior varies in hierarchical structure and provides a foundation for future work exploring broader cognitive capacities.

Reviewer comment 13: *When the authors generate random sequences for comparison, they draw integers from a uniform distribution. However, this choice may be problematic: ideally, the probabilities of drawing each integer in the simulated sequences should reflect the empirical frequencies observed in the chimpanzee nut-cracking and human stone-tool-making datasets.*

Author response: We thank the reviewer for this insightful suggestion. In response, we have revised our simulation approach to better reflect the empirical distribution of behaviors. Specifically, we calculated the probability of each behavior based on its overall frequency in the original dataset, and for each observed sequence, we generated a random control sequence of identical length by sampling from this empirical probability distribution. Although this adjustment did not substantially change the results (probably because there is not much variability in the probability of each behaviour, with empirical probabilities generally ranging between 0.2-0.4), we believe it has improved the robustness of our analyses because the credible intervals around our parameter estimates for the random group have generally become narrower, indicating greater precision in the estimates. Figure 4 has been updated accordingly, and the relevant section of the Results has been revised to reflect the new model outputs.

Reviewer comment 14: *It seems that the algorithm identifies exact repetitions of action sequences, such as [place nut][strike nut][eat nuts]. However, depending on task conditions, functionally equivalent behaviors may differ slightly in their execution. For instance, if a nut is unusually hard or a strike is missed, the sequence might appear as [place nut][strike nut][strike nut][eat nuts]. While this variation preserves the underlying structure, the algorithm would treat it as a distinct sequence due to the additional action, potentially missing structural similarities. This potential limitation does not appear to be directly addressed in the current manuscript.*

Author response: We thank the reviewer for another insightful comment. Our current analysis is indeed focused on the structural organization of action sequences, rather than inferring functional equivalence. While we recognize that surface variability (e.g., an additional strike) might sometimes reflect the “same” underlying function, in the present context such repetitions may in fact be meaningful. For example, nut hardness and the risk of damaging the kernel vary across strikes, and individuals may use different techniques to solve these challenges. From this perspective, a sequence such as [place nut][strike nut][strike nut][eat nut] is not necessarily functionally equivalent to [place nut][strike nut][eat nut], but rather may represent an alternative strategy or adaptive response. Determining equivalence of function therefore requires careful contextual analysis (e.g., nut properties, tool properties, anvil properties) that goes beyond the scope of the present study, but we agree it is an important avenue for future work and we have added a new discussion paragraph to address this issue (p 30 line 7 – p30 line 22).

Reviewer comment 15: *Briefly clarify what the ‘markovchain’ function does. I assume it fits the data to estimate the transition matrix, but it would be helpful for a broader audience if this were stated explicitly.*

Author response: We thank the reviewer for this helpful suggestion. We have clarified the role of the `markovchain` function in the Methods section (p8 line 13 – p8 line 17), noting that it estimates the

transition matrix by fitting the observed sequences to a Markov chain model. This should make the procedure clearer to a broader audience.

Reviewer comment 16: *The meaning of "the phrase length of each rule" is unclear for me — I had assumed that each rule was of length 2. Please clarify what is meant by this expression.*

Author response: We agree this was not clear and we have modified the sentence to clarify the meaning. The length of rule is indeed 2. However, one of those two characters may itself be another rule, meaning some rules have a length exceeding this (p9 line 16 – p9 line 19).

Reviewer comment 17: *The definitions of the measures of complexity are difficult to understand. It would be helpful if the authors could provide a clearer explanation of how these measures are defined and the rationale behind their definition.*

Author response: We agree that it was unclear, and we have modified table 1 to include both clearer definitions, and a rationale for each measurement, which we feel has greatly improved the readability of our methods section.

Reviewer comment 18: *The section "Analyzing hierarchical complexity" would benefit from greater clarity and precision. For example, the authors could provide more explicit definitions of the dependent and explanatory variables used in their models and clarify the interpretation of the alpha value.*

Author response: We agree that this aspect needed improvement, so we have now re-written our definitions and also provided rationales for the measurements in table 1.

Reviewer comment 19:

Remove "our" (p.3, l.7)

Add "7)" before "Tool change" and move to a new line (p.5, l.13)

Author response: Corrected.